# Preparation of High-Performance Barium Titanate Composite Hydrogels by Deep Eutectic Solvent-Assisted Frontal Polymerization

**DOI:** 10.3390/ma17133262

**Published:** 2024-07-02

**Authors:** Bin Li, Aolin Wu, Mengjing Zhou, Ying Wang, Zhigang Hu, Lihua Su

**Affiliations:** School of Mechanical Engineering, Wuhan Polytechnic University, Wuhan 430023, China

**Keywords:** deep eutectic solvent, frontal polymerization, hydrogel, barium titanate nanoparticles, piezoelectricity

## Abstract

This study aimed to develop composite hydrogels with exceptional piezoelectric properties and pressure sensitivity. To achieve the objective, this study created a deep eutectic solvent (DES) by mixing choline chloride (ChCl), acrylamide (AM), and acrylic acid (AA). Barium titanate nanoparticles (BTNPs) were incorporated as fillers into the deep eutectic solvents (DES) to synthesize the composite hydrogels using frontal polymerization (FP). The mechanical and piezoelectric properties of the resulting composite hydrogels were analyzed using Fourier transform infrared spectroscopy (FTIR) and scanning electron microscopy (SEM). This study found that the BTNPs/P(AM-co-AA) composite hydrogels exhibited excellent mechanical and piezoelectric properties. This is attributed to the high dielectric constant of BTNPs and the electrode polarization phenomenon when subjected to pressure. With a BTNPs content of 0.6 wt%, the maximum compressive strength increased by 3.68 times compared with the hydrogel without added BTNPs. Moreover, increasing the BTNPs content to 0.6 wt% resulted in a 1.48 times increase in generated voltage under the same pressure, compared with the hydrogel with only 0.2 wt% BTNPs. This study provides a method for preparing composite hydrogels with outstanding piezoelectric properties and pressure sensitivity.

## 1. Introduction

Hydrogel is a polymer widely utilized because it absorbs significant quantities of water without dissolving, owing to its cross-linked structure [1]. It is an environmentally sensitive hydrogel because external factors like light, pressure, temperature, and pH can cause changes in its internal network structure, solubility, strength, and other properties, leading to corresponding phase transitions. This gives it a certain level of intelligence and multifunctionality [2]. Hydrogel has the potential to be used in many fields, including drug release control [3], tissue engineering [4], sensors [5], and food engineering [6].

Frontal polymerization (FP) is a chemical process where monomers undergo localized reactions to create polymers at ambient temperature [7]. This polymerization method is highly efficient, fast, and energy saving, with rapid and uniform conversion of the monomer into polymer due to the exothermic reaction. The approach necessitates a small amount of organic solvents, is user friendly, and eliminates the need for energy-intensive or time-consuming separation procedures, decreasing environmental pollution. It has gained widespread popularity in the business [8]. Compared with traditional polymerization processes, FP offers several advantages, including a shorter reaction time, lower energy consumption, less emissions, and improved product characteristics. These benefits make it a promising option for experimental investigations and industrial production [9]. Organic solvents with a high boiling point, like dimethyl sulfoxide (DMSO), are typically needed for FP reactions. However, using toxic organic solvents might pose a risk to experimenters and lead to many issues, including contamination of the environment and other issues [10].

Barium titanate nanoparticles (BTNPs) are ceramic materials with attractive piezoelectric and dielectric properties that make them useful for bone engineering applications [11]. At the nanoscale, the properties of all materials inherently diverge from their bulk-scale counterparts, and barium titanate (BaTiO_3_) is no exception. The material’s innate high dielectric constant, ferroelectricity, and piezoelectricity observed at the macroscopic scale are significantly modulated by the crystallite dimensions at the nanoscale, an occurrence referred to as the size effect [12]. These unique properties are widely used in nonlinear optics [13], gate dielectrics [14], sensors [15], nonvolatile random access memory, and other applications. As the material’s size is reduced to the nanoscale, the dipole length in BaTiO_3_’s crystal structure is significantly affected, resulting in its exceptional dielectric and ferroelectric properties [16]. Their nanomorphic form’s microstructure, composition, applied stresses, defect concentration, and surface composition also contribute to their unique properties.

Deep eutectic solvents (DES) are substances that resemble ionic liquids and are created by combining donors of hydrogen bonds and acceptors. Chlorine chloride is the primary raw material used as a hydrogen bond acceptor in the production of DES, mainly due to its low cost and large production. DES based on choline chloride is a feasible alternative to ionic liquids since it has many advantages, including low cost, low volatility, non-toxicity, non-flammability, biodegradability, recyclability, and re-usability [17]. Over the years, DES has been increasingly used in various fields as a new, environmentally friendly, and non-toxic solvent and catalyst.

In this paper, BTNPs/P(AM-co-AA) composite hydrogels were prepared by a frontal polymerization method by introducing barium titanate nanoparticles (BaTiO_3_, BTNPs) into DES consisting of acrylamide, urea, and choline chloride (molar ratio of 1:1:1). The composite hydrogels were evaluated using scanning electron microscopy (SEM) and Fourier transform infrared spectroscopy (FTIR), and the effect of different contents of BTNPs on the hydrogel properties was explored. Compared with previous research work, in this paper, BTNPs were introduced as functional nanofillers, and their multi-faceted effects on hydrogel properties were explored, focusing on the effect of BTNPs content on the electrical properties of hydrogels and revealing the potential of nanomaterials in optimizing the capacitance and voltage of hydrogels. This study provides a new perspective on the application of nanomaterials in hydrogels and opens up new avenues for future hydrogel research directions and practical applications.

## 2. Experimentation

### 2.1. Materials

Choline chloride (ChCl), acrylic acid (AA), acrylamide (AM), potassium persulfate (KPS), and barium titanate nanoparticles (BTNPs) were purchased from Shanghai Aladdin Biochemical Science and Technology Co., Ltd. (Shanghai, China). The BTNPs reagent contained 99.9% of the main ingredient and the particle size was less than 100 nm. N,N-methylene bisacrylamide (MBA) was purchased from Tianjin Kemo Chemical Reagent Ltd. (Tianjin, China) and can be used directly. Before usage, it is necessary to subject ChCl to vacuum drying at a temperature of 70 °C for two hours to eliminate any absorbed water. All the reagents used have a high purity level, meeting analytical grade standards.

### 2.2. DES Preparation

The BTNPs were combined in the amounts shown in Table 1 after AM and AA as HBD and ChCl as HBA were combined in a beaker at a molar ratio of 1:1:1. The beaker was submerged in an oil bath and stirred manually with the oil level above the highest surface of the combined reagents using a collector-style thermostatically heated magnetic stirrer until a clear, cleared liquid materialized. After letting the prepared DES rest for a while, various concentrations of BTNPs were added to Table 1 and stirred until everything was homogenous. Figure 1 displays a diagram illustrating the internal hydrogen bonding structure of DES.

### 2.3. Preparation of Barium Titanate Composite Hydrogel

First, 0.5 wt% cross-linker (MBA) and 0.15 wt% initiator (KPS) were added to the combined DES and BTNP solution. This was mixed well, and transferred to a glass test tube (size 100 × 12 mm). The test tube was filled to approximately 80 mm above the bottom, and this was left to stand for a while to remove air bubbles created during stirring. A heated soldering iron initiated the upper liquid surface and removed once a stable front had formed and the polymerization reaction was initiated. The prepared hydrogel was removed from the reaction, sectioned into discs measuring 1–3 mm thick, submerged in deionized water for one week to remove unreacted monomers, and dehydrated at room temperature to constant weight before being set aside. The flow chart for the preparation of BTNPs/P(AM-co-AA) composite hydrogels is shown in Figure 2.

### 2.4. Performance Testing and Characterization of Composite Hydrogels

#### 2.4.1. SEM Characterization

The vacuum freeze-dried specimens were taken, and their cross sections were coated with gold using a high-vacuum ion sputtering machine. The cross-sectional morphology was subsequently examined under a scanning electron microscope.

#### 2.4.2. FTIR Characterization

The hydrogel was dried and pulverized into fine powder. The powder was then mixed with potassium bromide, powdered, pressed, and subjected to FTIR testing. In addition, the liquid sample was coated and also subjected to FTIR testing.

#### 2.4.3. Frontal Velocity and Frontal Temperature of Hydrogels

The frontal polymerization reaction often exhibits a consistent and constant speed of movement, which can be evaluated by analyzing the correlation between the change in frontal position and time. Before the reaction, the K-type thermocouple is inserted into the liquid surface at 70 mm below. The top of the liquid surface is triggered by the soldering iron, and, after the frontal front is formed, the temperature measured by the thermocouple is observed and recorded, and the relationship curve between the frontal position and the temperature can be obtained by using the obtained temperature.

#### 2.4.4. Piezoelectric Properties Testing of Hydrogels

A multimeter is used to measure the voltage on both sides of a cylindrical hydrogel with a 5 mm radius and a 10 mm thickness after being subjected to different pressures.

#### 2.4.5. Mechanical Testing of Hydrogels

The tensile properties of the composite hydrogel samples were verified on a micro-computer-controlled electronic universal testing equipment with a tensile speed of 100 mm/min until the maximal tensile stress was achieved after the samples were removed. The compressive properties of the composite hydrogel were tested on a TA.XTC-18 (Shanghai Baosheng Industrial Development Co., Ltd., Shangai, China) mass tester, and the compression experiments were carried out with the compression head at a speed of 0.2 mm/s until the maximum compressive stress was obtained.

The formulae for calculating the tensile and compressive strength of the material are as follows:(1)$$P=\frac{F}{S}$$
where *F* is the applied force and *S* is the surface area of the hydrogel.

#### 2.4.6. Hydrogel Swelling Property Test

A hydrogel weighing roughly 20 mg was placed in distilled water, and it was then periodically taken out, the water stains on its surface were drained with filter paper, and the weighing was monitored until the hydrogel’s weight stopped fluctuating. Equation (2) was then used to determine the hydrogel’s equilibrium swelling rate (SR):(2)$$SR=\frac{W_{t}-W_{0}}{W_{0}}$$

Wherein, *W*_0_ is the initial weight of the hydrogel and *W_t_* is the weight of the hydrogel after water absorption time *t*.

#### 2.4.7. Pressure Sensing Testing of Hydrogels

The composite hydrogel was prepared in a mold as a film of 2 cm × 2 cm in size, and the capacitance of the composite hydrogel was tested using a TH2827C precision LCR digital bridge (Suzhou New Tonghui Electronics Co., Ltd., Suzhou, China) Weights were used to apply pressure to the composite hydrogel, and, in order to produce an obvious deformation of the composite hydrogel after being subjected to pressure, 30 × 30 tiny protrusions were prepared on the surface of the composite hydrogel film using a mold to make the capacitance change more obvious. The obtained experimental data were fitted.

## 3. Results and Analysis

### 3.1. Frontal Speed and Temperature Measurements

During FP, a clear boundary known as the reaction front is formed between the polymer created through the reaction and the mixture of monomers that have not yet reacted [7]. Figure 3 shows the variation of frontal position and frontal temperature during frontal polymerization for DES with different contents of BTNPs added. As BTNPs content increases in the composite hydrogel, as Figure 3a illustrates, the reaction front’s movement decreases. This is because BTNPs, as a kind of polymer material, increase the viscosity of the reaction system after adding DES. The heightened viscosity leads to heightened resistance for the polymer during polymerization, leading to a decrease in the velocity of the frontal polymerization process. The frontal temperature versus time curve in Figure 3b indicates that the temperature stays relatively constant for the first two minutes of the reaction. Reaction temperature rises quickly when reaction time reaches two minutes and drops gradually when it reaches its maximum temperature. It indicates that the front peak is reacting downward at a constant rate. According to the phenomenon mentioned above, frontal polymerization is not caused by the monomer’s intrinsic spontaneous polymerization but rather by a heat source above the test tube [18].

### 3.2. Microscopic Morphology Analysis of Composite Hydrogels

Scanning electron microscopy was used to examine the hydrogel’s microscopic structure and determine how the BTNPs content affected the hydrogel’s morphology. As shown in Figure 4, the composite hydrogel had different microscopic morphologies with different BTNPs content. The figure indicates that when BTNPs were not added, the composite hydrogel had a porous and loose surface with a large pore size. However, an increased BTNPs content made the composite hydrogel’s surface denser, and the pore size gradually decreased. The formation of the porous structure may be related to high polymerization temperature and a rapid temperature increase. The high reaction temperature can vaporize solvents and monomers in the system, decompose the initiator, and form bubbles. After the bubbles are formed, they are captured by the viscous reaction system, creating pores. Increased BTNPs content formed tighter cross-linked structures in the hydrogels [19]. This phenomenon could be attributed to the increased surface area to volume ratio of BTNPs, which facilitates their interaction with the hydrogel polymer chains.

### 3.3. Fourier Infrared Spectral Analysis of Composite Hydrogels

Figure 5 displays the infrared spectra of the composite hydrogels, both with and without the inclusion of BTNPs. The absorption peak at 1455 cm^−1^ in the composite hydrogel, in the absence of BTNPs, is a result of the shearing of CH_2_ groups [20]. The absorption peaks at 2929 cm^−1^ and 2962 cm^−1^ result from the stretching of CH_2_ groups in AA and AM, respectively. The absorption peak at 3448 cm^−1^ is caused by the O-H groups in AA [21,22]. Incorporating BTNPs into the composite hydrogels results in an absorption peak at 586 cm^−1^ in the FTIR spectrogram, which can be attributed to the vibration of Ba-O in BTNPs [19]. In contrast, the absorption peak at 631 cm^−1^ can be attributed to Ti-O vibration in BTNPs [23]. Ti-O vibration in BTNPs, on the other hand, is responsible for the absorption peak at 631 cm^−1^ [23]. The findings above demonstrate the successful incorporation of BTNPs into the composite hydrogel.

### 3.4. Swelling Properties of Composite Hydrogels

Figure 6 displays the composite hydrogels’ swelling kinetic curves in deionized water. It is evident from Figure 6 that, after increasing the amount of BTNPs in the composite hydrogels from 0 weight percent to 0.6 weight percent, the four groups of composite hydrogels reached the swelling equilibrium in approximately 40 min, with only a slight variation in the time required to reach the swelling equilibrium. Nevertheless, the composite hydrogels experienced a gradual reduction in equilibrium swelling as the BTNP content increased. Specifically, the equilibrium swelling of FP1–FP4 was measured at 12.42, 11.02, 10.3, and 9, respectively. BTNPs exhibit hydrophobic characteristics. These nanoparticles may create microenvironments in hydrogels that change the hydrophilic equilibrium of the polymer chain segments. This can impact how nearby water molecules are arranged and how hydrogen bonds are formed between polymer chain segments [24]. At the same time, if the BTNPs aggregate within the hydrogel, this could result in an uneven internal structure, which would impact the hydrogel’s ability to transfer water molecules [25]. In the SEM images of composite hydrogels, we found that the presence of BTNPs increases the cross-link density of hydrogels, affecting the equilibrium swelling of hydrogels.

### 3.5. Mechanical Properties of Composite Hydrogels

The tensile and compressive stress–strain curves of the hydrogels are illustrated in Figure 7. Based on the data in Figure 7a, the FP1~FP4 hydrogels exhibited maximum tensile stresses of 11.73 MPa, 13.88 MPa, 14.93 MPa, and 20.56 MPa, respectively. The tensile strength of the composite hydrogels was enhanced as the content of BTNPs increased, and the highest tensile stresses of the FP4 hydrogels were 1.8 times greater than those of the FP1 samples. The compressive strength of the composite hydrogel increased gradually as the amount of BTNPs increased, as shown in Figure 7b. The maximum compressive strength of the FP4 hydrogel was 131.09 MPa, 3.68 times greater than that of FP1 (which had a maximum compressive strength of 35.57 MPa). The enhanced mechanical properties of the composite hydrogel can be attributed to the role of barium titanate nanoparticles as cross-linking agents, reinforcing the three-dimensional network structure of the hydrogel [26]. This cross-linking action improves the hydrogel’s stability and, as a result, the composite hydrogel’s mechanical qualities.

### 3.6. Pressure-Sensitive Features of Composite Hydrogels

Figure 8 displays pressure sensors’ capacitance from BTNPs/P(AM-co-AA) composite hydrogels at varying pressures. The pressure sensor’s capacitance increases progressively as the applied pressure increases, as illustrated in Figure 8. This is owing to the composite hydrogel’s extrusion and deformation after being subjected to an external force, which causes changes in the dielectric constant and electrode spacing, changing the capacitor’s electric field distribution and, as a result, its capacitance value. The pressure sensor’s capacitance increases with the amount of BTNPs it contains when exposed to the same pressure, as seen by the slope of the line in Figure 8. Because BTNPs have a high dielectric constant, the electric field may be focused more effectively when the dielectric layer has a higher dielectric constant [27]. When external pressure is applied to the capacitor, it changes the structure, thus affecting the distribution of the electric field and the capacitance value. When external pressure is applied to the capacitor, it causes a change in the capacitor’s structure, leading to an alteration in the distribution of the electric field and, consequently, the capacitance value [28]. Due to these properties, the composite hydrogel exhibits significant potential for application in health monitoring systems and smart wearables.

### 3.7. Piezoelectric Properties of Composite Hydrogels

Figure 9 illustrates a progressive increase in the produced voltage of the composite hydrogel as the applied pressure on the hydrogel increases. This is because the pressure deforms the barium titanate in the composite hydrogel, and the dipoles within the hydrogel are aligned in the direction of the external stimulus. Consequently, a potential difference between the upper and lower surfaces of the hydrogel is generated, resulting in an electric field within the hydrogel [29]. More dipoles in the composite hydrogel result in a larger potential differential between the top and lower surfaces and a higher voltage created as the barium titanate concentration of the hydrogel gradually increases. Therefore, it can be observed from Figure 9 that, with the increase in the content of barium titanate nanoparticles, the voltage generated from FP2 to FP4 gradually increases when subjected to the same pressure. Due to their distinctive physicochemical characteristics, BTNPs/P(AM-co-AA) composite hydrogels are mostly utilized in energy conversion and sensing applications. For instance, it may function as a sensor in intelligent materials to detect and respond to various stimuli such as temperature, pH, electric field, and more. Alternatively, its electrochemical characteristics can effectively convert and store energy in energy-harvesting devices.

## 4. Conclusions

This study reports on preparing BTNPs/P(AM-co-AA) composite hydrogels employing MBA as the cross-linking agent and KPS as the initiator. BTNPs were added as fillers into ChCl-AA-AM ternary DES. An experimental study was conducted to examine the composition and characteristics of the composite hydrogels:(1)The mechanical characteristics of the composite hydrogels were improved as the content of BTNPs increased, as BTNPs served as cross-linking sites in the hydrogels. Upon reaching a content of 0.6 wt% of BTNPs, the hydrogel exhibited a significant increase in its maximum tensile strength, approximately 1.8 times higher than that of the hydrogel without additional BTNPs. Similarly, the hydrogel’s maximum compressive strength rose by a factor of 3.68 compared with the hydrogel without BTNPs.(2)Due to BTNPs’ high dielectric constant, the BTNPs/P(AM-co-AA) composite hydrogel exhibits enhanced pressure sensitivity when the capacitor is fabricated as a dielectric layer. As the concentration of BTNPs in the composite hydrogel grows incrementally, the capacitor’s pressure sensitivity likewise increases progressively.(3)BTNPs are a common piezoelectric substance that confers piezoelectric capabilities to the composite hydrogel when applied as a filler. With an increase in the quantity of BTNPs in the composite hydrogel from 0.2 wt% to 0.6 wt%, the voltage produced by the composite hydrogel also rose from 44.8 mV to 63.2 mV when exposed to a pressure of 1200 Kpa.

## Figures and Tables

**Figure 1 materials-17-03262-f001:**
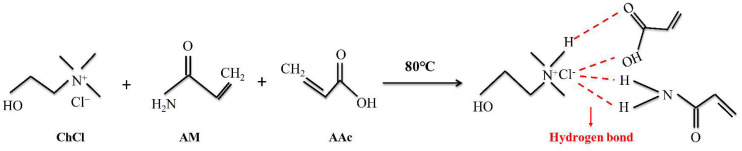
Formation equation of DES.

**Figure 2 materials-17-03262-f002:**
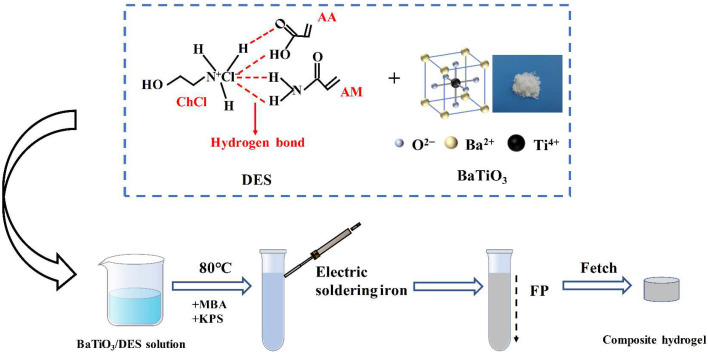
Principle of BTNPs/P(AM-co-AA) composite hydrogel preparation.

**Figure 3 materials-17-03262-f003:**
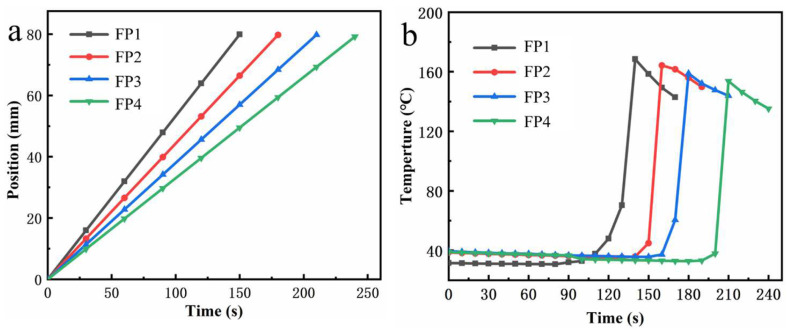
(**a**) shows the hydrogel’s front end’s position over time; (**b**) shows the hydrogel’s front end’s temperature over time.

**Figure 4 materials-17-03262-f004:**
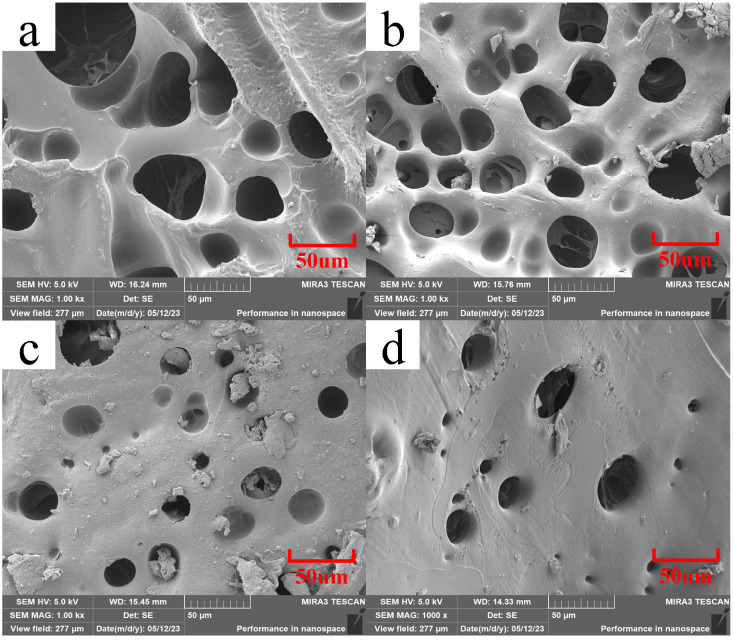
SEM images of FP1 (**a**), FP2 (**b**), FP3 (**c**), and FP4 (**d**) hydrogels.

**Figure 5 materials-17-03262-f005:**
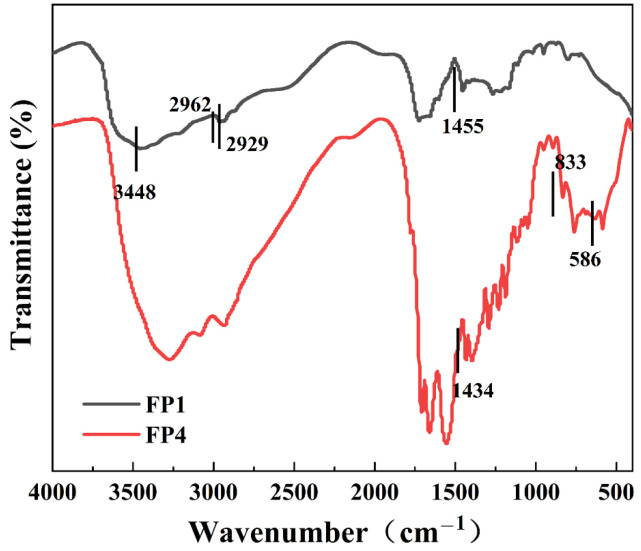
Fourier infrared spectra of FP1 and FP4.

**Figure 6 materials-17-03262-f006:**
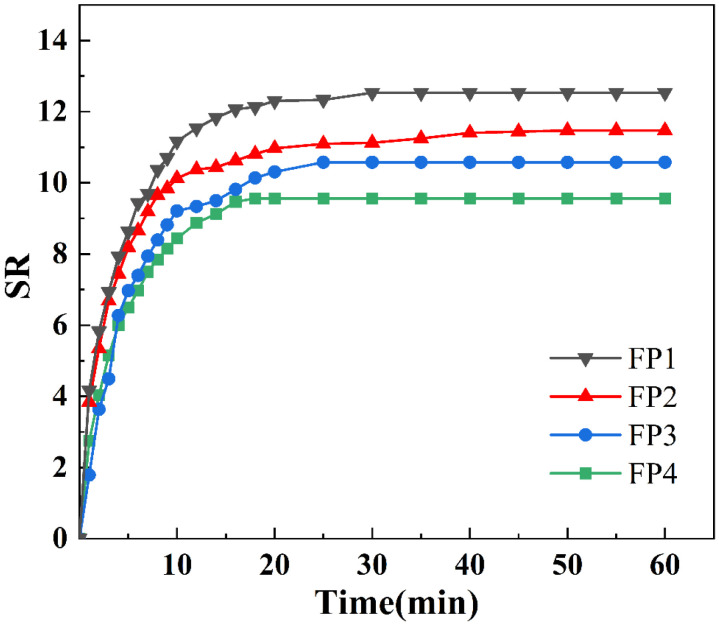
Swelling kinetics curves of composite hydrogels.

**Figure 7 materials-17-03262-f007:**
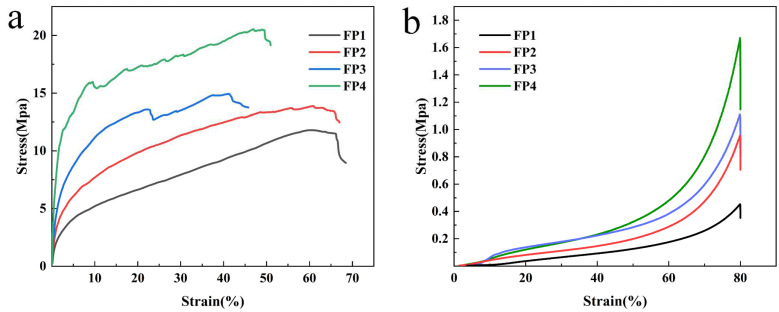
(**a**) Hydrogel tensile performance test curve; (**b**) hydrogel compression performance test curve.

**Figure 8 materials-17-03262-f008:**
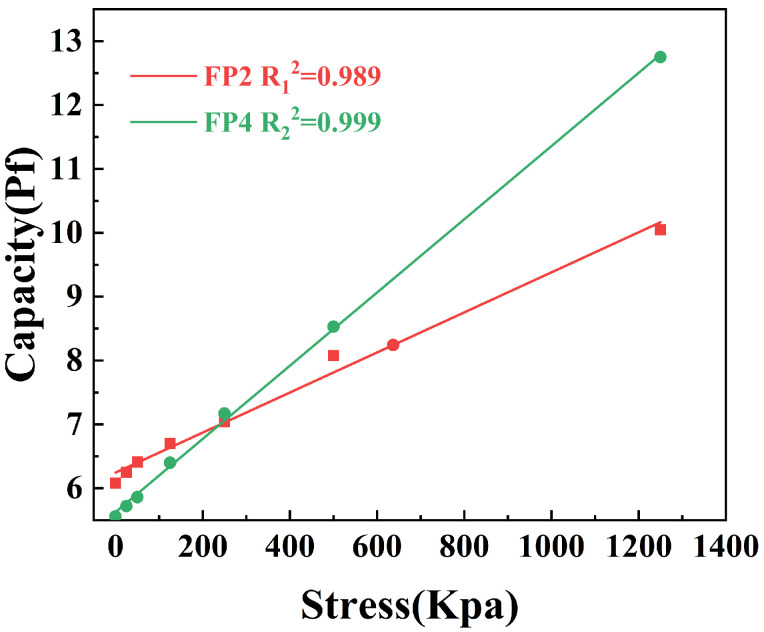
Change in capacitance of a pressure sensor when subjected to different pressures.

**Figure 9 materials-17-03262-f009:**
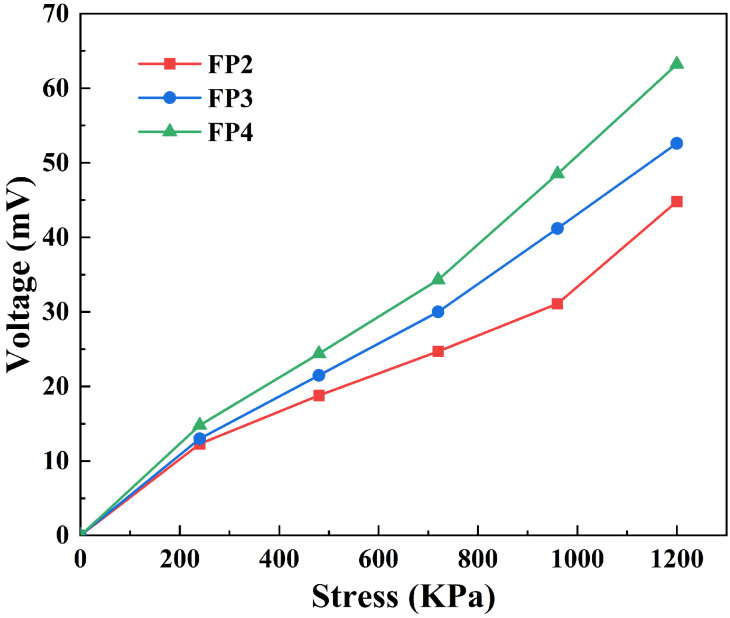
Piezoelectric performance curves of composite hydrogels subjected to different pressures.

**Table 1 materials-17-03262-t001:** Composition and proportion of DES.

Sample	AA:AM:ChCl (Molar Ratio)	BTNPs (wt%)	KPS(g) (wt%)	MBA(g) (wt%)
FP1	1:1:1	0	0.15	0.5
FP2	1:1:1	0.2	0.15	0.5
FP3	1:1:1	0.4	0.15	0.5
FP4	1:1:1	0.6	0.15	0.5

## Data Availability

The original contributions presented in the study are included in the article, further inquiries can be directed to the corresponding author.

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
