# Peer review of "Preparation of High-Performance Barium Titanate Composite Hydrogels by Deep Eutectic Solvent-Assisted Frontal Polymerization"

_materials, 2024, doi:10.3390/ma17133262_

Round 1

Reviewer 1 Report

Comments and Suggestions for Authors

Title: Preparation of BTNPs/P(AM-co-AA) composite hydrogels by front-end polymerization of ternary DES and its performance study

This original article presents the study of a composite hydrogel formulation using BTNPs/P(AM-co-AA) in which physical and mechanical parameters are studied. Overall, the manuscript is well organized and makes correct use of English. In addition, the results are accompanied by figures and images that facilitate understanding. However, there are some considerations that authors should consider. The suggestions are shown below:

0.       Abstract, title and keywords

The abstract provides the most relevant information of the manuscript, and it is in general well-written. Some considerations are given:

·         Line 10: authors should consider changing “they” and write a passive voice sentence.

The title is accurate and in concordance with the manuscript topic.

The keywords are well-selected.

1.       Introduction

The introduction is well-written and organized. Moreover, it provides general and important information to understand the topic of the manuscript. However, some suggestions are written below for the manuscript quality improvement:

·         Line 50: What do the authors mean by “Although all materials behave differently at the nanoscale than in bulk, BaTiO3 50 is no exception”? That BaTiO3 50 is not an exception or that it is?

·         Authors should consider closing this section with some lines explaining the goal of the manuscript (line 67).

2.       Material and methods

This section is well-organized in subsections and provides fundamental information to understand the formation and study of the hydrogels.

3.       Result and discussion

This section is well-organized in subsections and provides a well-discussed results that are complemented with figures and graphics, which helps to understand the manuscript.

However, there are some points that could be improved:

·         Section 3.1: frontal polymerization has already been written as FP in the Introduction section. It is not necessary to repeat it.

·         Figure 3: it appears above some numbers that do not correspond with the graphic.

·         Lines 166-167: Is the statement “The high viscosity causes the polymer to encounter higher resistance during polymerization, leading to a decrease in front-end speed” based on any previous publication, or is it a conclusion reached by the authors?

·         Section 3.3: Why did the authors choose only FP1 and FP4 for the Fourier infrared spectra?

·         Figure 5 includes “FP0” but there is no FP0 described before in Table 1.

·         Lines 243-246: authors should rewrite this sentence to make it easier for the reader to understand the statement.

·         Section 3.6: Why do the authors study the pressure-sensitive properties only in FP2 and FP4?

·         In graphics, it is suggested to use the same colors for the same hydrogels, e.g., FP2 always red.

4.       Conclusions

The conclusions are written in such a way that the main results obtained after the formation of the hydrogels are recorded. Moreover, they are presented in a clear and orderly manner, making them easy to understand. However, the authors may consider incorporating future studies to complement the information already obtained.

Final marks and conclusions

In my opinion, this original article provides relevant information for the composite hydrogel formulation, and it is an interesting topic with great potential for industrial applications. However, there are some key points that authors should change to understand the information properly. Therefore, I suggest MINOR REVISIONS.

Comments on the Quality of English Language

Authors should rewrite some sentences.

Overall the quality is good. 

Reviewer 2 Report

Comments and Suggestions for Authors

I have reviewed the submitted manuscript on “Preparation of BTNPs/P(AM-co-AA) composite hydrogels by front-end polymerization of ternary DES and its performance study” by Bin Li et al submitted to Materials. Compared with traditional polymerization methods, the reported frontal polymerization has the advantages of short reaction time while having low reaction time and low pollution. These excellent characteristics make it potentially promising for experimental research and industrial production.

In the submitted work, the authors have used deep eutectic solvent a new type of ionic liquid, and obtained copolymer hydrogels while adding Ch-Cl. FP is a matured area in polymers however, adding barium titanate as filler into the polymer would bring some insights to the current well-known work. This is a significant area of research, however, the work requires a lot of clarity. The work in its present form is not suitable and needs some revisions before rendering a final decision.

My specific points are below

·         Better not to include too many acronyms in the title.

·         What is the rationale for choosing BTNP as filler?

·         In the abstract, line 9, “researchers created..” it should be authors in the current work created…”

·         How the current work vary from that of the author’s recently reported work in the literature New Journal of Chemistry apart from adding BTNPs? This may be lacking novelty unless otherwise clearly stated.

·         The clarity of the scale in the Fig 4 SEM images must be improved.

·         How does the BTNP cross-links with the hydrogels?

·         In Figure 5, the O-H and CH2 group spectra must refer to the similar IR bands reported in the literature such as (10.1039/D3DT03736C).

·         The term ESR and how it influences the hydrogens must be concluded in section 3.4.

·         Section 3.6, line 296, is this subjected to different pressure or "weight"?

·    The structure of BTNPs must be given in the earlier part of the manuscript.

·         Reference 26, has not been provided with doi.

·         Reference 9; RSC  (SC capital letter).

The application of the current reported material must be emphasized in the results and d discussion section.

Round 2

Reviewer 2 Report

Comments and Suggestions for Authors

 The authors have performed some minor changes but the manuscript is still written very loose. The revised version is not meeting the journal requirements. Please see below.

·         The authors must articulate clearly in the manuscript how the current works differentiate from that of the author recently reported in the literature New J. Chem.

·         The clarity of the scale bar in Fig. 4 is still poor.

·         Why the FT-IR has been shown only for FP1 and FP4?

·         In Fig.6, y-axis unit what does SR stand for?

·         Section 3.6 – it says in a change in capacitance but in Fig. 8 y-axis it says capacity. Which is correct?

·         What are the primary applications for the reported composite hydrogels? Although it is stated in line 271 (p. 9) it is unclear.

Comments on the Quality of English Language

Marginal, the revised parts need to be substantially improved.
